# The anthropometric and physical qualities of women's rugby league Super League and international players; identifying differences in playing position and level

Sean Scantlebury[1,2]*, Sam McCormack[1,2], Thomas Sawczuk[1,2,3], Stacey Emmonds[1,2], Neil Collins[1,2], Jake Beech[1], Carlos Ramirez[1,2], Cameron Owen[1,4], Ben Jones[1,2,5,6,7]

1 Carnegie Applied Rugby Research (CARR) Centre, Carnegie School of Sport, Leeds Beckett University, Leeds, United Kingdom, 2 England Performance Unit, Rugby Football League Ltd, Leeds, United Kingdom, 3 School of Built Environment, Engineering and Computing, Leeds Beckett University, Leeds, United Kingdom, 4 Leeds Rhinos Netball, Leeds, United Kingdom, 5 Leeds Rhinos Rugby League Club, Leeds, United Kingdom, 6 Division of Exercise Science and Sports Medicine, Faculty of Health Sciences, University of Cape Town, Cape Town, South Africa, 7 School of Science and Technology, University of New England, Armidale, New South Wales, Australia

* s.scantlebury@leedsbeckett.ac.uk

**Data Availability Statement:** All relevant data are within the paper and its Supporting information files.

## Abstract

Participation in women's rugby league has been growing since the foundation of the English women's rugby league Super League in 2017. However, the evidence base to inform women's rugby league remains sparse. This study provides the largest quantification of anthropometric and physical qualities of women's rugby league players to date, identifying differences between positions (forwards & backs) and playing level (Women's Super League [WSL] vs. International). The height, weight, body composition, lower body strength, jump height, speed and aerobic capacity of 207 players were quantified during the pre-season period. Linear mixed models and effects sizes were used to determine differences between positions and levels. Forwards were significantly ($p < 0.05$) heavier (forwards: 82.5 ± 14.8kg; backs: 67.7 ± 9.2kg) and have a greater body fat % (forwards: 37.7 ± 6.9%; backs: 30.4 ± 6.3%) than backs. Backs had significantly greater lower body power measured via jump height (forwards: 23.5 ± 4.4cm; backs: 27.6 ± 4.9cm), speed over 10m (forwards: 2.12 ± 0.14s; backs: 1.98 ± 0.11s), 20m (forwards: 3.71 ± 0.27s; backs: 3.46 ± 0.20s), 30m (forwards: 5.29 ± 0.41s; backs: 4.90 ± 0.33s), 40m (forwards: 6.91 ± 0.61s; backs: 6.33 ± 0.46s) and aerobic capacity (forwards: 453.4 ± 258.8m; backs: 665.0 ± 298.2m) than forwards. Additionally, international players were found to have greater anthropometric and physical qualities in comparison to their WSL counterparts. This study adds to the limited evidence base surrounding the anthropometric and physical qualities of elite women's rugby league players. Comparative values for anthropometric and physical qualities are provided which practitioners may use to evaluate the strengths and weaknesses of players, informing training programs to prepare players for the demands of women's rugby league.

**Funding:** The funder (the Rugby Football League) provided support in the form of salaries for authors [SS, SM, CR, BJ], but did not have any additional role in the study design, data collection and analysis, decision to publish, or preparation of the manuscript. The specific roles of these authors are articulated in the 'author contributions' section.

**Competing interests:** The authors would like to declare a commercial affiliation with the Rugby Football League. This affiliation does not alter our adherence to PLOS ONE policies on sharing data and materials.

## Introduction

Participation in women's rugby league is increasing [1]. The number of Australian women playing rugby league in 2018 increased by 29%, whilst participation in the UK has increased linearly since 2015 with a 35% growth in school programs from 2015 to 2019 [2]. However, despite continuous growth, research within women's rugby league is sparse, with a recent call to action [1] highlighting the need to increase the evidence base within the sport.

Rugby league is an intermittent collision sport comprised of intense activities (e.g. sprinting, tackling) interspersed with bouts of lower intensity activity (e.g. walking) [3]. Due to the demanding nature of rugby league, players require a range of well-developed anthropometric and physical qualities (e.g. speed, power, body composition) to meet game demands, optimise performance and reduce the likelihood of injury [4, 5]. Therefore, the anthropometric and physical qualities of female rugby league players have received inceptive attention [6]. Initial research found female Australian international backs to be quicker than forwards over 10 (backs: 1.96 ± 0.10s; forwards: 2.04 ± 0.10s), 20 (backs: 3.44 ± 0.14s; forwards: 3.60 ± 0.19s) and 40 meters (backs: 6.33 ± 0.25s; forwards: 6.59 ± 0.25s), have greater muscular power (backs: 35.7 ± 5.9cm; forwards: 35.1 ± 8.0cm), agility (backs: 2.64 ± 0.19s; forwards: 2.63 ± 0.13s) and estimated maximal aerobic power (backs: 32.2 ± 4.4ml·kg$^{-1}$·min$^{-1}$; forwards: 35.3 ± 3.43ml·kg$^{-1}$·min$^{-1}$). On the other hand, forwards were heavier (forwards: 75.5 ± 12.5kg; backs: 64.7 ± 7.6kg) with a greater sum of seven skinfolds (forwards: 141.2 ± 37.2mm; backs: 114.8 ± 20.2mm) in comparison to backs [6].

Jones et al., [7] found English international representative backs to be quicker than forwards over 10m (backs: 1.87 ± 0.09s; forwards: 2.01s ± 0.17s), 20m (backs: 3.36 ± 0.18s; forwards: 3.60s ± 0.26s), 30m (backs: 4.68 ± 0.25s; forwards: 5.05 ± 0.44s) and 40m (backs: 6.13 ± 0.25s; forwards: 6.59s ± 0.61s). Furthermore, backs had greater agility turning off their right (backs: 2.59 ± 0.11s; forwards: 2.70 ± 0.15s) and left foot (backs: 2.58 ± 0.14s; forwards: 2.74 ± 0.21s), and greater power (measured via a countermovement jump; backs: 0.29 ± 0.05m; forwards: 0.24 ± 0.05m). Forwards had a greater body mass (backs: 66.0 ± 7.3kg; forwards: 80.7 ± 14.3kg) and percentage body fat (backs: 27.7 ± 4.8%; forwards: 33.5 ± 5.6%) compared to backs. These findings substantiate the earlier work of Gabbett [6] who investigated Australian international women's rugby league players.

Whilst the previous work of Gabbett [6] and Jones et al., [7] provides an initial insight into the anthropometric and physical characteristics of elite women's rugby league players, sports such as rugby union and soccer have demonstrated an increase in physical qualities over time with players becoming stronger, faster and fitter, in line with the increased professionalism of the game [8, 9]. Consequently, further research is required to assess the impact of the increased exposure, participation and organisation of the women's game (e.g., inception of the English Women's Super League [WSL] in 2017). Furthermore, the existing literature quantifying the anthropometric and physical characteristics of women's rugby league players have concentrated on small samples (n = 32, [6], n = 27, [7]) of international level players. Previous literature in male rugby league has shown physiological characteristics to differentiate between playing levels [10, 11]. A larger sample size comprising of international and non-international women's rugby league players is required to develop a holistic quantification of anthropometric and physical qualities and evaluate any differences which may exist between levels of competition (i.e., international vs non-international).

This study aims to increase the evidence base in women's rugby league by quantifying the anthropometric (height, body mass, body composition) and physical (strength, power, speed, aerobic capacity) qualities of female rugby league players and identifying any differences that may exist between international and Women's Super League (WSL) players.

## Materials and methods

### Participants

A total of 207 women's rugby league players from all 10 WSL clubs in England (100 forwards [age 23.2 ± 5.8]; 82 backs [age 21.5 ± 4.8]) and the England international side (12 forwards [age 23.7 ± 4.0]; 13 backs [age 23.8 ± 4.8]) were tested during the 2019 pre-season period. Due to factors such as equipment failure and adverse weather conditions, not every participant recorded a score for each test. Table 1 displays the number of participants who recorded a score for each test for each combination of playing position and level. Written consent was provided by all of the WSL clubs as well as the national side. All testing procedures were clearly explained prior to testing. Ethics for the experimental procedures were granted prior to data collection by Leeds Beckett University (ethical clearance number: 69658).

### Design of study

The testing battery was designed to quantify standing height, body mass, body composition (bioelectrical impedance analysis), lower body muscular power via jump height (countermovement jump [CMJ]), muscular strength (isometric mid-thigh pull [IMTP]), speed (10, 20, 30, 40m sprint) and aerobic capacity (modified Yo-Yo intermittent recovery fitness test level 1 [modified Yo-Yo IRT1]). Standing height, body mass, body composition, muscular power and strength tests were completed indoors before moving outdoors to complete speed and aerobic capacity tests. Outdoor tests were completed on either a grass or artificial surface. The constraints of the testing battery were to ensure that all players within a squad (n = ~20) could be tested within a single session (typically 1 hour). All testing was completed by the research team, visiting each club to ensure standardisation during the pre-season period. Prior to testing, participants were asked to provide information regarding their date of birth and typical playing position and performed a standardised warm up. Participants completed anthropometric, CMJ, muscular strength and speed testing prior to the modified Yo-Yo IRT1. Two trials were conducted for muscular power, muscular strength and speed testing, with the participants' best score recorded.

### Procedures

**Anthropometrics and body composition.** Standing height was measured to the nearest 0.1cm using a portable stadiometer (Seca 213, Hamburg, Germany). Body mass was collected using calibrated analogue scales (Seca, Hamburg, Germany) to the nearest 0.1 kg. Bioelectrical impedance analysis (Tanita BF-350, Tokyo, Japan) was used to quantify body fat percentage. Previous research has demonstrated bioelectrical impedance analysis to have excellent reliability with a test re-test interclass correlation coefficient (ICC) of 0.98 [12].

**Muscular strength.** To assess muscular strength, the IMTP was performed using a dynamometer (T.K.K.5402, Takei Scientific Instruments Co. Ltd, Niigata, Japan) sampling at 122

**Table 1. The number of participants who completed each test for each combination of playing level and position.**

|  | Height | Body Mass | Body Fat % | CMJ | IMTP | 10m | 20m | 30m | 40m | Modified Yo-Yo IRT1 |
|---|---|---|---|---|---|---|---|---|---|---|
| WSL Forwards | 94 | 94 | 87 | 92 | 91 | 95 | 95 | 95 | 75 | 94 |
| WSL Backs | 78 | 79 | 71 | 74 | 74 | 80 | 80 | 80 | 66 | 78 |
| International Forwards | 12 | 12 | 12 | 12 | 12 | 11 | 11 | 11 | 11 | 11 |
| International Backs | 13 | 13 | 13 | 13 | 13 | 13 | 13 | 13 | 13 | 12 |

WSL = Women's Super League. CMJ = Countermovement Jump. IMTP = Isometric Mid-Thigh Pull, IRT1 = Intermittent Recovery Test Level 1

Hz, which was attached to a wooden platform, a chain and a latissimus pulldown bar. The test protocol outlined by Till et al., [13] was utilised in which participants were positioned by standing with their feet approximately shoulder width apart with the chain length adjusted so that the bar was positioned at the mid-thigh. Participants were instructed to maintain a flat back position with their head up and arms straight. Subjects gripped the bar, maintaining tension in the chain prior to beginning the pull, to ensure a jerk action was not performed. Participants pulled directly upwards, keeping their feet flat on the floor and without leaning back. The highest dynamometer score of the two attempts was recorded in kilograms. Despite a slight underestimation, a strong significant relationship has been demonstrated between the peak force derived from a dynamometer and that of a force platform (r = 0.92, P<0.001) [14] subsequently indicating appropriate construct validity in a cohort of senior and youth professional rugby league players. Furthermore, the dynamometer has been shown to have acceptable between day reliability (TE as CV = 5.5% [4.5–6.9]) [15].

**Lower body muscular power.** Lower body muscular power was assessed via jump height using a CMJ. The CMJ was performed on two portable force plates (PS-2141, Pasco, Roseville, California, USA). Participants began with their legs fully extended with their hands on their hips. The depth of the countermovement was self-selected with no attempt made to control the depth or speed of the countermovement. Participants were instructed to keep their legs extended in flight and to land with their legs straight. Previous research has found portable force plates to be reliable when quantifying CMJ height with an ICC and coefficient of variation (CV) for CMJ height of 0.85 and 3.8% respectively [16].

**Speed.** Speed was evaluated over 10, 20, 30 and 40m using photocell timing gates (Brower Timing Systems, Salt Lake City, UT). Participants started in their own time, 0.5m (marked with a cone) behind the first gate in a 2-point stance. Two maximal efforts were performed with a 3-minute rest separating each trial. Previous research has found Brower timing systems to be reliable when quantifying 10, 20, 30 and 40m sprints with mean typical errors expressed as a coefficient of variation of 2.5%, 2.2%, 2.2% and 1.8% respectively [15]. Furthermore, the validity of Brower timing systems to asses maximum velocity has been established in comparison to the criterion measure of a radar gun with a small typical error of estimate (1.67% [1.46–1.97]) and nearly perfect correlation (r = 0.97 [0.95–0.98]) [17].

**Aerobic capacity.** A modified version on the prone Yo-Yo IRT1 was utilised to quantify aerobic capacity. The modified Yo-Yo IRT1 required participants to complete 2 x 15m shuttle runs, interspersed with 10 seconds of active recovery in which participants were required to walk to and from a cone placed 5m behind the start line. Participants started each stage of the test in prone position with their chest flat to the floor, legs straight and head behind the start line. The speed of the shuttles increased as the test progressed and is controlled by audio signals dictating the time in which shuttles need to be completed within. The speed of the test increased progressively with the players stopping of their own volition or until they had failed to meet the start/finish line in the allocated time, two times. The concurrent validity of the 20m prone Yo-Yo IRT1 has been previously established in male academy rugby league players [18] however the present study reduced the shuttle distance to 15m to account for the physiological differences between male and female athletes [6, 7, 10, 19].

## Statistical analysis

To evaluate the differences between anthropometric and physical qualities, linear mixed models were used. Each anthropometric (height, body mass, body composition) and physical (strength, power, speed, aerobic capacity) quality was added to its own model as the dependent variable. A fully factorial model was produced, whereby position, playing level and the

position*playing level interaction were included as fixed effects. Club was included as a random effect to account for any clustering in anthropometric and physical qualities that could occur due to coach selection priorities or training schedules. Pairwise differences were used to evaluate the differences in the least square means between position (forwards vs backs), playing level (international vs WSL) and the position*playing level interaction (every combination of position and playing level). Statistical significance was set at P<0.05. Cohen's *d* effect sizes were used to establish the magnitude of difference, thresholds were set as: 0.2 *small*, 0.6 *moderate*, 1.2 *large*, 2.0 *very large*. Data were analysed using SAS University Edition (SAS Institute, Cary, NC).

## Results

The height, body mass and body fat % for all WSL forwards and backs and international forwards and backs are presented in Fig 1. The CMJ, IMTP and modified Yo-Yo IRT1 values are presented in Fig 2, with 10, 20, 30 and 40m sprint times for WSL and international forwards and backs presented in Fig 3. The mean ± standard deviation for all anthropometric and physical qualities for forwards and backs combined, international & WSL players combined and international and WSL forwards and backs considered separately are presented in Table 2. Table 3 displays the effect sizes, 95% confidence intervals and P values for differences between international and WSL forwards and backs.

### Anthropometric characteristics

There was a *moderate* and significant difference in height between international and WSL players with international players taller due to a *large* significant difference in height between international backs and WSL backs. A *large* and significant difference in body mass was found with forwards heavier than backs. *Large* and significant differences were present as WSL forwards had a greater body mass than WSL and international backs, with a *moderate* and significant difference between international forwards and WSL backs. Forwards had a *large* and significantly higher body fat % than backs with *large* and significant differences between WSL forwards and WSL and international backs and a *moderate* significant difference between international forwards and WSL backs.

### Physical qualities

*Large* and significant differences were found in jump height with backs jumping higher than forwards and international players jumping higher than WSL players. There was a *large* and significant difference between forwards with international forwards jumping higher than WSL forwards. International backs had a greater jump height compared to WSL forwards, international forwards and WSL backs with all differences *large* and significant. WSL forwards had a *small* and significantly higher IMTP score than WSL backs.

*Moderate* and *large* significant differences in sprint times were found with backs quicker than forwards and international players quicker than WSL players over 10m, 20m, 30m, and 40m. There were *moderate*, *large* and significant differences as international backs had quicker sprint times than international forwards over 10 and 20m, WSL forwards over 10, 20, 30 and 40m and WSL backs over 10, 20 and 30m. WSL backs had quicker sprint times than WSL forwards over 10, 20, 30 and 40m with all differences *large* and significant. The sprint times for international forwards were quicker than WSL forwards with *large* and significant differences over 10, 20, 30 and 40m.

*Moderate* and significant differences were found between backs and forwards and international and WSL players for the modified Yo-Yo IRT1 with backs completing more meters than

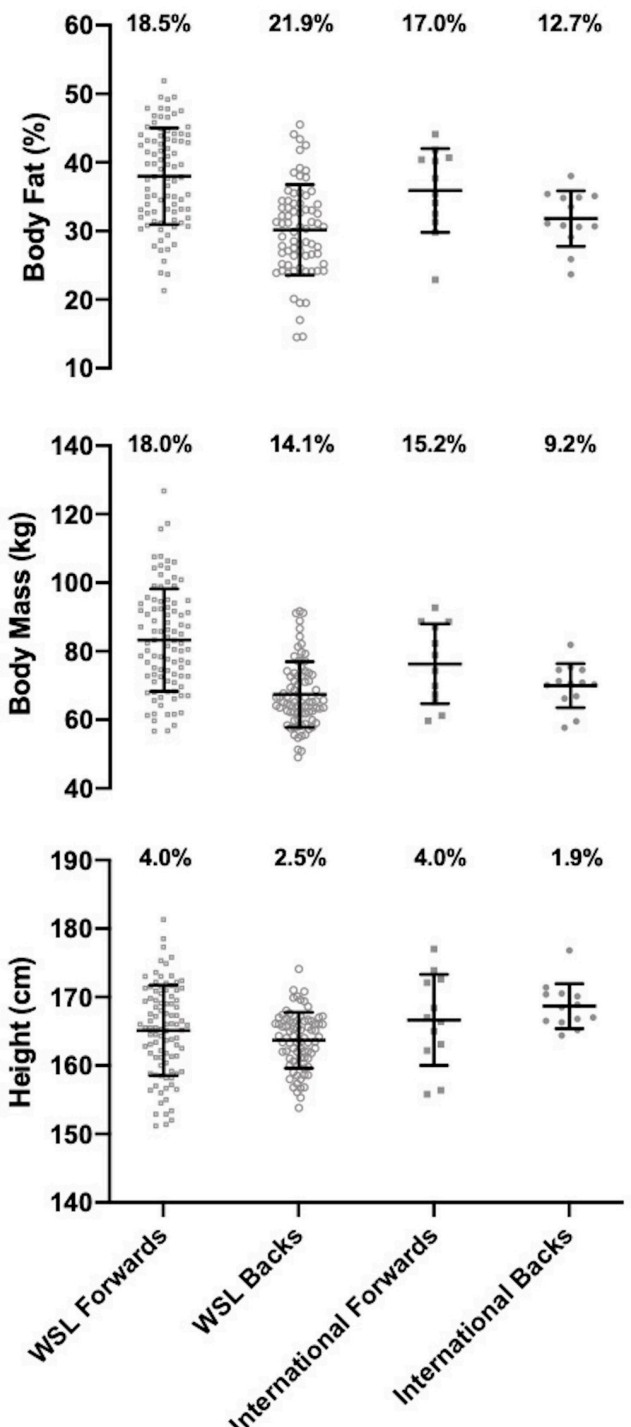

**Fig 1. The height, body mass and body fat % for WSL forwards and backs and international forwards and backs.**

forwards and international players completing more meters than WSL players. There was a *large* and significant difference between international backs and WSL forwards with international backs completing more meters. International forwards and WSL backs completed more meters than WSL forwards with differences *moderate* and significant.

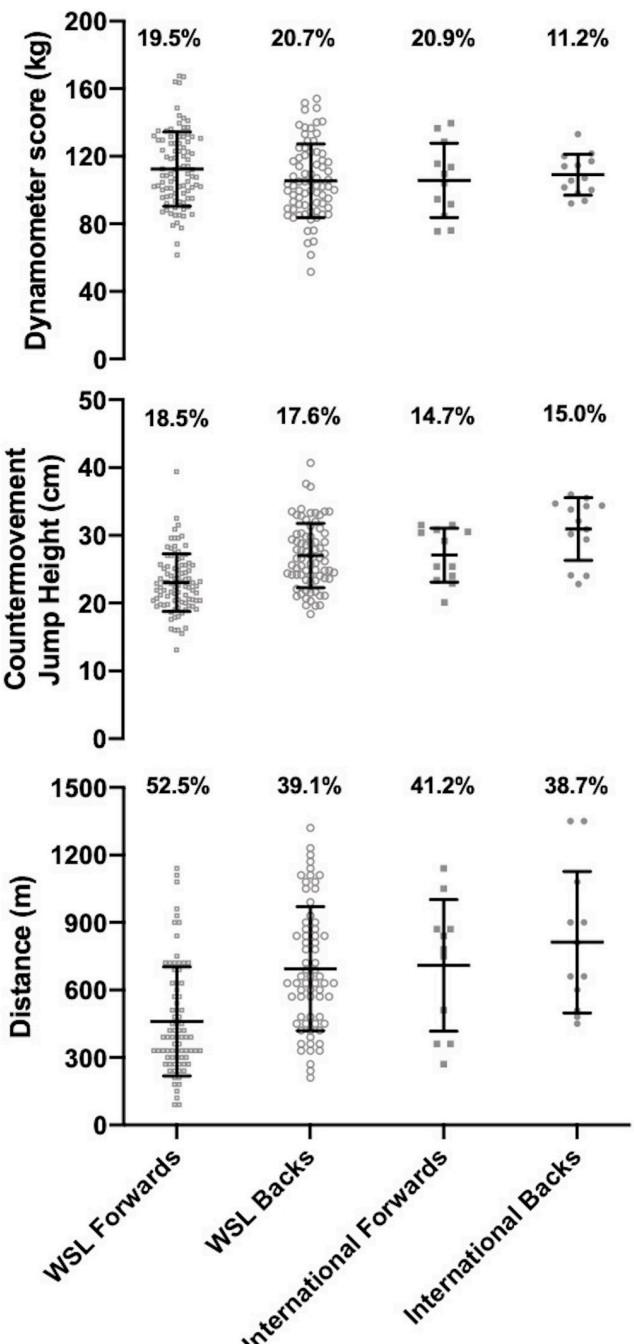

**Fig 2. The CMJ, IMTP and modified Yo-Yo IRT1 scores for WSL forwards and backs and international forwards and backs.**

## Discussion

This study aimed to quantify the anthropometric and physical qualities of female rugby league players using the largest sample size on this cohort to date, identifying differences between international and WSL players and playing positional groups. Findings of this study substantiate previous literature in female rugby league [6, 7] with forwards found to be heavier than

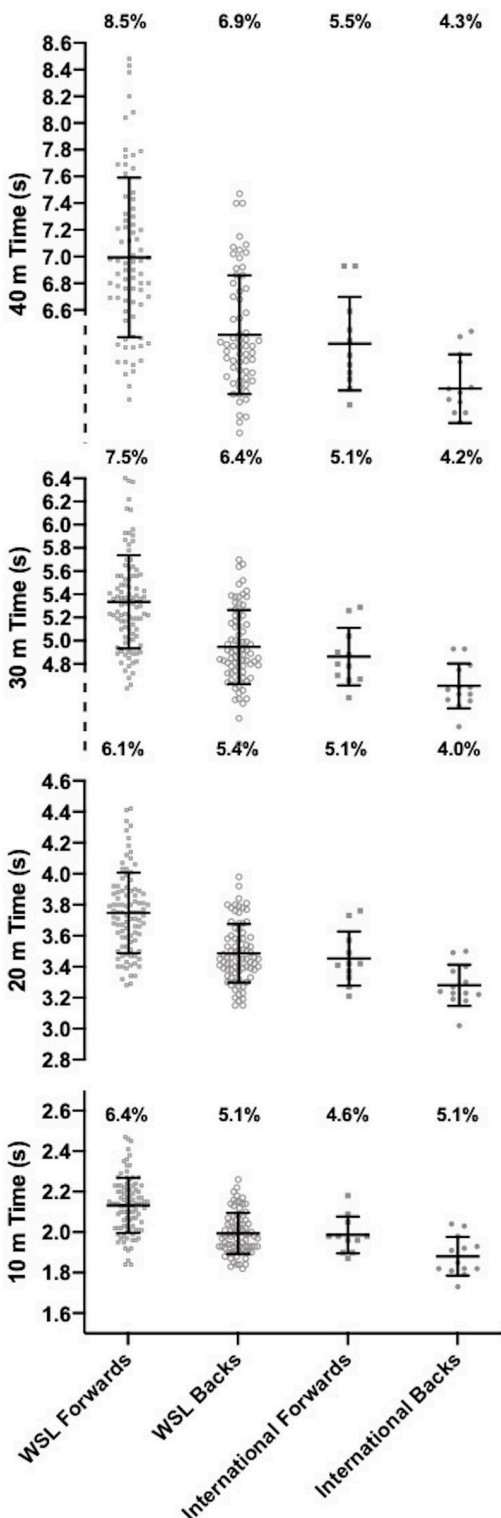

**Fig 3. The 10, 20, 30, and 40m times for WSL forwards and backs and international forwards and backs.**

**Table 2. Anthropometric and physical qualities (mean ± SD) for WSL forwards and backs and international forwards and backs.**

| | Height (cm) | Body Mass (kg) | Body Fat (%) | CMJ (cm) | IMTP (kg) | 10m (s) | 20m (s) | 30m (s) | 40m (s) | Modified Yo-Yo IRT1 (m) |
|---|---|---|---|---|---|---|---|---|---|---|
| Forwards | 165.3 ± 6.6 | 82.5 ± 14.8 | 37.7 ± 6.9 | 23.5 ± 4.4 | 111.6 ± 21.9 | 2.12 ± 0.14 | 3.72 ± 0.27 | 5.29 ± 0.41 | 6.91 ± 0.61 | 453.4 ± 258.8 |
| Backs | 164.4 ± 4.3 | 67.7 ± 9.2 | 30.4 ± 6.3 | 27.6 ± 4.9 | 106.0 ± 20.6 | 1.98 ± 0.11 | 3.46 ± 0.20 | 4.90 ± 0.33 | 6.34 ± 0.44 | 665.0 ± 298.2 |
| WSL | 164.5 ± 5.6 | 76.0 ± 15.0 | 34.5 ± 7.9 | 24.8 ± 4.9 | 109.3 ± 22.1 | 2.07 ± 0.14 | 3.63 ± 0.26 | 5.16 ± 0.41 | 6.71 ± 0.61 | 522.7 ± 284.8 |
| International | 167.7 ± 5.2 | 73.0 ± 9.6 | 33.8 ± 5.4 | 29.1 ± 4.7 | 107.5 ± 17.3 | 1.93 ± 0.11 | 3.36 ± 0.17 | 4.73 ± 0.25 | 6.17 ± 0.34 | 763.0 ± 301.9 |
| WSL Forwards | 165.1 ± 6.6 | 83.3 ± 15.0 | 38.0 ± 7.0 | 23.0 ± 4.3 | 112.4 ± 21.9 | 2.13 ± 0.14 | 3.75 ± 0.26 | 5.34 ± 0.40 | 6.99 ± 0.60 | 423.5 ± 238.9 |
| WSL Backs | 163.7 ± 4.1 | 67.4 ± 9.6 | 30.2 ± 6.6 | 27.0 ± 4.8 | 105.5 ± 21.8 | 1.99 ± 0.10 | 3.49 ± 0.19 | 4.95 ± 0.32 | 6.41 ± 0.44 | 642.3 ± 291.0 |
| International Forwards | 166.7 ± 6.6 | 76.3 ± 11.6 | 35.9 ± 6.1 | 27.1 ± 4.0 | 105.8 ± 22.1 | 1.99 ± 0.09 | 3.45 ± 0.17 | 4.86 ± 0.25 | 6.35 ± 0.35 | 709.1 ± 292.2 |
| International Backs | 168.7 ± 3.3 | 70.0 ± 6.4 | 31.8 ± 4.0 | 30.9 ± 4.6 | 109.0 ± 12.2 | 1.88 ± 0.10 | 3.28 ± 0.13 | 4.61 ± 0.19 | 6.01 ± 0.26 | 812.5 ± 314.8 |

**Table 3. The differences (effect size, 95% CI and P value) between WSL forwards and backs and international forwards and backs for anthropometric and physical quality measures.**

| | Height | Body Mass | Body Composition | CMJ | IMTP | 10m | 20m | 30m | 40m | Modified Yo-Yo IRT1 |
|---|---|---|---|---|---|---|---|---|---|---|
| Forwards *vs.* | -0.03 | 0.91* | 0.92* | 0.94* | 0.12 | 0.98* | 0.96* | 0.88* | 0.89* | -0.58* |
| | (-0.46 to 0.40) | (0.48 to 1.34) | (0.50 to 1.34) | (1.37 to 0.51) | (-0.31 to 0.54) | (0.59 to 1.37) | (0.58 to 1.34) | (0.51 to 1.26) | (0.53 to 1.26) | (-0.98 to -0.18) |
| Backs | p = 0.88 | p = 0.00 | p = 0.00 | p = 0.00 | p = 0.59 | p = 0.00 | p = 0.00 | p = 0.00 | p = 0.00 | p = 0.01 |
| International *vs.* | -0.57* | 0.20 | 0.11 | 0.96* | 0.10 | 0.92* | 0.90* | 0.83* | 0.61* | -0.51* |
| | (-1.00 to 0.14) | (-0.26 to 0.67) | (-0.37 to 0.60) | (1.43 to 0.48) | (-0.38 to 0.57) | (0.46 to 1.38) | (0.35 to 1.25) | (0.39 to 1.27) | (0.16 to 1.07) | (-0.99 to -0.03) |
| WSL | p = 0.01 | p = 0.38 | p = 0.64 | p = 0.00 | p = 0.68 | p = 0.00 | p = 0.00 | p = 0.00 | p = 0.01 | p = 0.04 |
| International Forwards *vs.* | -0.28 | 0.57 | 0.40 | 0.92* | 0.31 | 1.13* | 1.02* | 1.05* | 0.89* | -0.71* |
| | (-0.88 to 0.33) | (-0.07 to 1.21) | (-0.25 to 1.05) | (1.58 to 0.27) | (-0.34 to 0.96) | (0.49 to 1.76) | (0.40 to 1.63) | (0.44 to 1.66) | (0.27 to 1.50) | (-0.99 to -0.03) |
| WSL Forwards | p = 0.37 | p = 0.10 | p = 0.23 | p = 0.01 | p = 0.34 | p = 0.00 | p = 0.00 | p = 0.00 | p = 0.01 | p = 0.03 |
| WSL Forwards *vs.* | 0.26 | 1.28* | 1.20* | 0.91* | 0.33* | 1.18* | 1.18* | 1.11* | 1.17* | -0.77* |
| | (-0.05 to 0.56) | (0.98 to 1.58) | (0.90 to 1.51) | (1.21 to -0*.60) | (0.03 to 0.63) | (0.92 to 1.45) | (0.92 to 1.43) | (0.85 to 1.36) | (0.90 to 1.44) | (1.04 to -0.51) |
| WSL Backs | p = 0.10 | p = 0.00 | p = 0.00 | p = 0.00 | p = 0.03 | p = 0.00 | p = 0.00 | p = 0.00 | p = 0.00 | p = 0.00 |
| International Backs *vs.* | -0.60 | 1.12* | 1.03* | 1.90* | 0.21 | 1.90* | 1.76* | 1.71* | 1.51* | -1.09* |
| | (-1.20 to 0.01) | (0.50 to 1.74) | (0.40 to 1.67) | (2.52 to 1.28) | (-0.40 to 0.83) | (1.33 to 2.47) | (1.20 to 2.32) | (1.16 to 2.26) | (0.95 to 2.06) | (-1.69 to -0.48) |
| WSL Forwards | p = 0.05 | p = 0.00 | p = 0.00 | p = 0.00 | p = 0.50 | p = 0.00 | p = 0.00 | p = 0.00 | p = 0.00 | p = 0.00 |
| International Forwards *vs.* | 0.53 | 0.71* | 0.81* | 0.02 | 0.02 | 0.06 | 0.16 | 0.05 | 0.28 | -0.07 |
| | (-0.08 to 1.15) | (0.06 to 1.35) | (0.15 to 1.46) | (-0.64 to 0.68) | (-0.63 to 0.67) | (-0.57 to 0.69) | (-0.45 to 0.77) | (-0.56 to 0.66) | (-0.33 to 0.89) | (-0.70 to 0.57) |
| WSL Backs | p = 0.09 | p = 0.03 | p = 0.02 | p = 0.96 | p = 0.96 | p = 0.85 | p = 0.61 | p = 0.90 | p = 0.40 | p = 0.84 |
| International Forwards *vs.* | -0.32 | 0.55 | 0.64 | 0.98* | -0.10 | 0.78* | 0.74* | 0.66 | 0.62 | -0.38 |
| | (-1.13 to 0.48) | (-0.26 to 1.36) | (-0.16 to 1.43) | (1.78 to 0.17) | (-0.89 to 0.69) | (0.05 to 1.51) | (0.03 to 1.45) | (-0.05 to 1.36) | (-0.06 to 1.30) | (-1.13 to 0.37) |
| International Backs | p = 0.43 | p = 0.18 | p = 0.11 | p = 0.02 | p = 0.81 | p = 0.04 | p = 0.04 | p = 0.07 | p = 0.10 | p = 0.32 |
| International Backs *vs.* | -0.86* | -0.16 | -0.17 | 0.99* | -0.12 | 0.72* | 0.58* | 0.61* | 0.34 | -0.31 |
| | (-1.47 to 0.24) | (-0.79 to 0.46) | (-0.81 to 0.47) | (1.62 to 0.36) | (-0.74 to 0.51) | (0.15 to 1.29) | (0.03 to 1.14) | (0.05 to 1.16) | (-0.21 to 0.89) | (-0.92 to 0.29) |
| WSL Backs | p = 0.01 | p = 0.61 | p = 0.60 | p = 0.00 | p = 0.71 | p = 0.01 | p = 0.04 | p = 0.03 | p = 0.23 | p = 0.31 |

*Denotes a statistically significant difference (p <0.05)

backs with a higher body fat %. On the other hand, backs had greater lower body power, speed over 10, 20, 30 and 40m and aerobic capacity. International players were taller than WSL players with greater lower body power, speed over 10, 20, 30 and 40m and aerobic capacity. These findings demonstrate the discrepancy in anthropometric and physical qualities between playing positions and playing levels.

Whilst the findings of this study largely support existing evidence [6, 7], differences can be seen when comparing within positions at the international level. Gabbett [6], Jones et al., [7] and the present study found the average 40m sprint time for backs to be 6.33s, 6.13s and 6.01s respectively. A similar trend is found over 20m and 30m with backs and forwards quicker in this study in comparison to previous literature [6, 7]. Additionally, international forwards and backs in this study had greater lower body power (measured via jump height) (forwards: 27.1 ± 4.0cm, backs: 30.9 ± 4.6cm) in comparison to international forwards and backs (forwards: 24.0 ± 0.1cm, backs: 29.0 ± 0.1cm) [7]. Improvements in speed and lower body power may be indicative of enhanced anthropometric and physical qualities due to an increase in the professionalism of women's rugby league alongside increased provision (e.g., access to strength and conditioning coaches). However, it cannot be conclusively stated that anthropometric and physical qualities have improved since initial research was conducted as comparisons between other anthropometric (e.g., body fat %) and physical (maximal strength) qualities are difficult to interpret accurately due to differences in the test and equipment used between studies. Future research should seek to keep the testing battery and testing equipment consistent to longitudinally assess changes in the anthropometric and physical qualities of women's rugby league players.

Improved anthropometric and physical qualities have previously been shown to positively influence playing level [11, 20]. Such findings are corroborated by the results of this study as international players were more powerful, faster and had greater aerobic capacity in comparison to their WSL counterparts. Greater anthropometric and physical qualities may contribute to international selection as the demands of rugby league require a range of well-developed anthropometric and physical characteristics [3]. However, once selected, international women's rugby league players have access to a greater level of provision (e.g., structured strength and conditioning training sessions) compared to non-selected players, which may increase the disparity in anthropometric and physical qualities.

Whilst previous literature has presented means and standard deviations for anthropometric and physical measures, the small sample sizes have prevented analysis of the variability within the dataset [6, 7]. Figs 1–3 present the coefficient of variation of anthropometric and physical qualities across each of the level and position combinations. Alongside the coefficient of variation, the visualisation of WSL forwards and backs data points displays the large variability in each of the anthropometric and physical measures. Large variability is prominent for measures of strength, jump height, aerobic capacity, body mass and body fat %. Such variability may be symptomatic of the different levels of provision available across the 10 WSL clubs. Discrepancies in provision include access to higher quality facilities (e.g. gyms) and the participation in structured strength and conditioning programs [21]. To elevate anthropometric and physical standards, all WSL clubs and players should be provided access to education regarding appropriate strength and conditioning practices. Due to the financial restrictions currently present in women's rugby league, it is not feasible for all WSL clubs to employ qualified strength and conditioning practitioners to administer and deliver strength and conditioning programs. Therefore, facilitating educational opportunities, such as strength and conditioning workshops, is a crucial first step in advancing the current knowledge base in women's rugby league.

To the authors knowledge, this study provides the largest quantification of anthropometric and physical qualities of women's international and Super League rugby league players.

Despite this, the study is not without limitation. The 10, 20, 30 and 40m sprints alongside the modified Yo-Yo IRT1 had to be completed outside, on a grass or an artificial surface. Subsequently, variations in weather conditions may have influenced test scores. Whilst scores affected by adverse weather conditions have been removed from analysis, the possibility of weather conditions impacting test performance cannot be entirely ruled out. Future research should attempt to keep testing conditions consistent between clubs (e.g., surface type) or, when this is not possible, ensure all participants are wearing appropriate footwear.

## Conclusion

The findings of this study update and substantiate previous literature quantifying the anthropometric and physical qualities of women's rugby league players whilst also comparing measures between playing levels. Backs and international players were found to have greater lower body power, speed and aerobic capacity in comparison to forwards and WSL players whilst forwards were found to be heavier with a higher body fat % in comparison to backs. Overall, this study provides position specific comparative data for the anthropometric and physical qualities of women's rugby league players. However, the variability in anthropometric and physical qualities for WSL players should be considered when evaluating mean values. Moving forward, focus should be placed on elevating anthropometric and physical qualities across the entirety of the WSL by increasing strength and conditioning provision and knowledge.

## Practical applications

Due to the demanding nature of rugby league match-play, players are required to have well developed anthropometric and physical qualities [3]. The importance of anthropometric and physical qualities is reinforced by their ability to enhance performance and reduce the risk of injury [10, 22]. Therefore, it is important to understand the current anthropometric and physical qualities of the highest level of women's rugby league in England, the women's rugby league Super League and the international squad. This study provides the largest quantification of anthropometric and physical qualities of women's rugby league players to date, offering generalisable position specific comparative values. Practitioners may utilise these values to analyse the strengths and weaknesses of their players in comparison to WSL and international level players. Such analysis may inform subsequent training programs to ensure players are prepared for the rigours of women's rugby league.

The large sample of WSL players analysed highlights the variability in anthropometric and physical qualities. The variability in anthropometric and physical qualities may be symptomatic of the varying levels of provision available to players. To elevate the anthropometric and physical standards across the WSL, players and clubs should be provided with access to education regarding appropriate strength and conditioning practises to increase the knowledge base and reduce discrepancies in anthropometric and physical qualities.

## Supporting information

**S1 Data. Minimal data set.**
(XLSX)

## Author Contributions

**Conceptualization:** Sean Scantlebury, Ben Jones.

**Data curation:** Sean Scantlebury, Sam McCormack, Stacey Emmonds, Neil Collins, Jake Beech, Carlos Ramirez.

**Formal analysis:** Thomas Sawczuk, Neil Collins.

**Methodology:** Sean Scantlebury, Thomas Sawczuk, Ben Jones.

**Writing – original draft:** Sean Scantlebury.

**Writing – review & editing:** Sean Scantlebury, Sam McCormack, Stacey Emmonds, Carlos Ramirez, Cameron Owen, Ben Jones.

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
