## [Decision Letter · Decision Letter 0]

25 Jun 2021

PONE-D-21-08836

The anthropometric and physical qualities of women’s rugby league Super League and international players; identifying differences in playing position and standard

PLOS ONE

Dear Dr. Scantlebury,

Thank you for submitting your manuscript to PLOS ONE. After careful consideration, we feel that it has merit but does not fully meet PLOS ONE’s publication criteria as it currently stands. Therefore, we invite you to submit a revised version of the manuscript that addresses the points raised during the review process.

We look forward to receiving your revised manuscript.

Kind regards,

Caroline Sunderland

Academic Editor

PLOS ONE

Journal Requirements:

'The authors have declared that no competing interests exist' 

We note that one or more of the authors are employed by a commercial company: Rugby Football League Ltd.

Additional Editor Comments (if provided):

Reviewers' comments:

Reviewer's Responses to Questions

**Comments to the Author**

1. Is the manuscript technically sound, and do the data support the conclusions?

Reviewer #1: Yes

Reviewer #2: Partly

2. Has the statistical analysis been performed appropriately and rigorously? 

Reviewer #1: Yes

Reviewer #2: Yes

3. Have the authors made all data underlying the findings in their manuscript fully available?

Reviewer #1: Yes

Reviewer #2: Yes

4. Is the manuscript presented in an intelligible fashion and written in standard English?

Reviewer #1: Yes

Reviewer #2: Yes

5. Review Comments to the Author

Reviewer #1: Overall, I think the goal of this study is good, and adds to the limited body of research in women’s team sports. I am unsure about the aim of this study addressing the “evidence gap” around the quantification of anthropometric and physical qualities of female rugby league players, though. I agree that there is limited research in women’s rugby league, but as pointed out in your introduction this isn’t the first study to look at anthropometric or physical qualities. Your paper does a good job at adding to the previous literature by increasing the sample size of the cohort and including comparisons between international and women’s super league (WSL), but provides me with little new information than what existing evidence already does to address this “evidence gap”.

The inclusion and explanation of all procedures performed in the battery testing are exceptional, and provide quality information for others trying to replicate these tests. However, you mention in your procedures for aerobic capacity that a reduced shuttle distance was used to account for physiological differences between men and women; has this reduction in shuttle distance been quantified previously, or did you use do any internal validation for this? Do you also have a reference highlighting previous men and women’s physiological differences?

In regard to general writing structure, the paper is well structured and written.

Reviewer #2: Revise the title as follows: Comparison of anthropometric and physical qualities of elite women’s rugby league players across playing level and position.

Abstract:

Replace playing standard with playing level.

Add the number of teams used in the data collection.

Introduction

The introduction is well written and leads the reader to the aim of the study.

Page 3: Line 68: Removed more recently.

Material and methods

Add the ethical clearance number and institution.

Add a note under Table 1 to explain the abbreviation used.

Can the results for the outdoor test be compared if you using different surfaces?

A major concern is the modified Yo-Yo test used to determine the aerobic capacity? Validity of the test? Jones et al 2016 made use of the 20m shuttle.

The results and discussion will be reviewed in the 2nd revision based on the feedback provided on the Yo-Yo test that tested the aerobic capacity.

6. PLOS authors have the option to publish the peer review history of their article (what does this mean?). If published, this will include your full peer review and any attached files.

Reviewer #1: **Yes: **Rebecca Peek

Reviewer #2: **Yes: **Wilbur Kraak

---

## [Author Response · Author response to Decision Letter 0]

28 Jul 2021

Thank you to both reviewers for your comments. We believe the suggested changes have improved the quality of the manuscript. 

A response to the reviewers document as well as a manuscript with tracked changes can be found towards the bottom of the document. 

Reviewer 1

Overall, I think the goal of this study is good, and adds to the limited body of research in women’s team sports. 

I am unsure about the aim of this study addressing the “evidence gap” around the quantification of anthropometric and physical qualities of female rugby league players, though. I agree that there is limited research in women’s rugby league, but as pointed out in your introduction this isn’t the first study to look at anthropometric or physical qualities. 

Your paper does a good job at adding to the previous literature by increasing the sample size of the cohort and including comparisons between international and women’s super league (WSL), but provides me with little new information than what existing evidence already does to address this “evidence gap”. 

• Thank you for your comments. We agree that ‘evidence gap’ may not be the correct terminology, therefore the line “This study aims to address this evidence gap” has been changed to “This study aims to increase the evidence base in women’s rugby league”.

The inclusion and explanation of all procedures performed in the battery testing are exceptional, and provide quality information for others trying to replicate these tests. 

However, you mention in your procedures for aerobic capacity that a reduced shuttle distance was used to account for physiological differences between men and women; has this reduction in shuttle distance been quantified previously, or did you use do any internal validation for this? 

• Thank you for your comment. This is an issue we have considered as a research group and appreciate your concerns. The prone Yo-Yo IRT1 was used following the validation of the protocol by Dobbin et al., 2021 (reference added below). The prone Yo-Yo IR1 was more strongly associated with common measures of rugby league training and match loads (table 1) than the Yo-Yo IR1 with the authors concluding that the prone Yo-Yo IR1 offers an appropriate measure of rugby-specific high intensity intermittent running that partially explains the changes in internal and external load during simulated rugby league match play.

Table 1: The relationship between the prone Yo-Yo IR1 and Yo-Yo IR1 and common measures of internal and external load following simulated rugby league match play

 Prone Yo-Yo IR1 Yo-Yo IR1

% Relative distance r = 0.61 r = 0.57

% Mean speed r = 0.64 r = 0.36

High metabolic power r = 0.48 r = 0.25

Fatigue index r = 0.71 r = 0.63

% HR peak r = -0.56 r = -0.35

RPE 1st half r = -0.44 r = -0.14

RPE 2nd half r = -0.68 r = -0.41

The 20m prone Yo-Yo IR1 was initially used in the testing battery, however, the 20m distance, alongside starting in the prone position was judged to be inappropriate for the cohort. This was because multiple participants were failing during the initial stages of the test which increased the homogeneity of the testing scores. The grouping of testing scores reduced the usefulness of scores to WSL clubs who utilised the testing results to differentiate the fitness levels of their players. Therefore, the decision was made to keep the prone element of the Yo-Yo IR1 test due to its increased validity to simulated rugby league match play but reduce the distance from 20m to 15m. From a practical perspective, the reduced distance increased the sensitivity of the testing measure facilitating a greater comparison of fitness levels. Whilst the authors appreciate that the 15m prone Yo-Yo has not been specifically validated via previous literature, we believe that the similarity in this protocol to the validated 20 prone Yo-Yo offers an appropriate measure of aerobic capacity and has been included within the testing battery. 

• Dobbin, N., Highton, J., Moss, S. L., Hunwicks, R., & Twist, C. (2021). Concurrent validity of a rugby-specific Yo-Yo intermittent recovery test (level 1) for assessing match-related running performance. The Journal of Strength & Conditioning Research, 35(1), 176-182.

Do you also have a reference highlighting previous men and women’s physiological differences?

Thank you for the comment, this is an omission that should have been placed in the initial submission. References have now been added which highlight differences in the fitness levels of male and female rugby league players.

Research by Gabbett et al., (2013), found semi-professional male rugby league players selected to start a representative match ran (mean ± SD) 1506 ± 338m in the Yo-Yo IR1. This is in comparison to Jones et al., (2016) who found international women’s rugby league players to complete 728 ± 154m during the Yo-Yo IR1.

Furthermore, Gabbett et al. (2007), found estimated VO2 max (ml·kg-1·min-1) scores to range from (mean ± SD) 46.9 ± 4.8 ml·kg-1·min-1, 45.6 ± 5.7 ml·kg-1·min-1, and 47.6 ± 7.6 ml·kg-1·min-1 for first, second and third grade players respectively following a multi-stage fitness test. Comparatively, international female rugby league players were found to have an estimated VO2 max (ml·kg-1·min-1) of 32.2 ± 4.4 ml·kg-1·min-1 and 35.3 ± 3.4 ml·kg-1·min-1 for forwards and backs respectively following a multi-stage fitness test (Gabbett, 2007)

• Gabbett, T. J., & Seibold, A. J. (2013). Relationship between tests of physical qualities, team selection, and physical match performance in semiprofessional rugby league players. The Journal of Strength & Conditioning Research, 27(12), 3259-3265.

• Gabbett, T. I. M., Kelly, J., & Pezet, T. (2007). Relationship between physical fitness and playing ability in rugby league players. The Journal of Strength & Conditioning Research, 21(4), 1126-1133.

• Gabbett, T. J. (2007). Physiological and anthropometric characteristics of elite women rugby league players. The Journal of Strength & Conditioning Research, 21(3), 875-881.

• Jones, B., Emmonds, S., Hind, K., Nicholson, G., Rutherford, Z., & Till, K. (2016). Physical qualities of international female rugby league players by playing position. The Journal of Strength & Conditioning Research, 30(5), 1333-1340.

Specific comments are provided below.

L101 Be consistent with how you describe playing standard. Previously and further into the paper you use the term international, yet here you say national side. 

• Thank you for your comment, this has now been changed to state international side.

L178 Do you have a reference for physiological differences between males and females.

• Thank you for your comment, the references have now been added.

L180 Statistical analysis: Where international players included in both competition levels (i.e. international and WSL), are the international group just a subsample of your overall dataset? if so, how did you account for this in your statistical analysis? Do you think this would affect your results? 

• Thank you for your comment, we coded international vs WSL (i.e. playing level) as a separate variable to club (i.e. the club the player played for). All players played in the WSL, but international players were coded as international, whereas non-international players were coded as WSL. As such, international players could be considered a subset of the data. To control for the similarities we'd expect players within each club to have regardless of their competition level (e.g., due to S&C practices), we used the club as a random effect. Consequently, the mean difference between international and WSL players accounts for the issue that you have highlighted.

Did you also consider having the individual athlete as a random effect to account for potential individual differences, not just positional?

• Thank you for your suggestion, however, we couldn't use player as random effect because each physical attribute was considered in individual models and each player only had one observation for each attribute. Therefore, there is a potential limitation that we don't understand the covariance between physical qualities.

Table 2. The first two rows in your table are forwards and backs, then followed by the competition level and positions within those competition levels. It is unclear what those first two positions are highlighting? Are they the combination of WSL and international players, or? And if so how is the average height of those forwards (row 1 in table) significantly lower than other forwards listed in the table? 

• Thank you for your comment and noticing this mistake. We agree that increased clarity was required. Therefore, a sentence has been added into the results section (lines 316 – 317) to explain the data presented in table two. The height for combined forwards within the table was a typo and has now been amended. 

Table 3. In the heading you include (int) I assume as an abbreviation for international but then don’t use it here in this table or anywhere else in previous tables or figures nor within text. Is it necessary?

• Thank you for your observation. This abbreviation has now been removed. 

Table 3. I was surprised by the inclusion of cross positional analysis across different competition levels (i.e. International backs v WSL forwards). From your statistical analysis section, you highlight position, playing standard and then position*playing standard as fixed effects, but it might be worth clearly stepping through and outlining all of the different analyses that you have completed to ensure you cover off the statistical approach for everything.

• Thank you for your suggestion. We appreciate additionally clarity was required in the statistical analysis section; therefore, we have added further information to highlight what each pairwise differences we outline was used to compare. In this case, position*playing level was used to compare between forwards and backs across both playing levels (Lines 189-192). 

Reviewer 2the title as follows: Comparison of anthropometric and physical qualities of elite women’s rugby league players across playing level and position

Abstract:

Replace playing standard with playing level.

• Thank you for your comment, this change has been made throughout the document.

Add the number of teams used in the data collection.

• Thank you for your comment, the number of WSL clubs (10) is included on line 106 in the participants section of the methodology. 

Introduction

The introduction is well written and leads the reader to the aim of the study.

Page 3: Line 68: Removed more recently.

• Thank you for your comment, this has now been removed. 

Material and methods

Add the ethical clearance number and institution.

• Thank you for your comment, this has now been added. 

Add a note under Table 1 to explain the abbreviation used.

• Thank you, abbreviations have now been added to underneath the table.

Can the results for the outdoor test be compared if you using different surfaces?

• Thank you for your comment, we agree that this is a limitation of the study and is incorporated within the study limitations paragraph of the discussion. However, due to the restricted facilities of the 10 WSL clubs it was not possible to standardise the testing surface. The decision was made to include both grass and artificial surface testing scores in the results to increase participants numbers, allowing for the quantification of the entirety of the women’s super league. 

A major concern is the modified Yo-Yo test used to determine the aerobic capacity? Validity of the test? Jones et al 2016 made use of the 20m shuttle.

• Thank you for your comment. This is an issue we have considered as a research group and appreciate your concerns. The prone Yo-Yo IRT1 was used rather than the Yo-Yo IRT 1 as used by Jones et al., 2016 following the validation of the prone Yo-Yo IR1 by Dobbin et al., 2021 (reference added below). The prone Yo-Yo IR1 was more strongly associated with common measures of rugby league training and match loads (table 1) than the Yo-Yo IR1 with the authors concluding that the prone Yo-Yo IR1 offers an appropriate measure of rugby-specific high intensity intermittent running that partially explains the changes in internal and external load during simulated rugby league match play.

Table 1: The relationship between the prone Yo-Yo IR1 and Yo-Yo IR1 and common measures of internal and external load following simulated rugby league match play

 Prone Yo-Yo IR1 Yo-Yo IR1

% relative distance r = 0.61 r = 0.57

% mean speed r = 0.64 r = 0.36

High metabolic power r = 0.48 r = 0.25

Fatigue index r = 0.71 r = 0.63

% HR peak r = -0.56 r = -0.35

RPE 1st half r = -0.44 r = -0.14

RPE 2nd half r = -0.68 r = -0.41

The 20m prone Yo-Yo IR1 was initially used in the testing battery, however, the 20m distance, alongside starting in the prone position was judged to be inappropriate for the cohort. This was because multiple participants were failing during the initial stages of the test which increased the homogeneity of the testing scores. The grouping of testing scores reduced the usefulness of scores to WSL clubs who utilised the testing results to differentiate the fitness levels of their players. Therefore, the decision was made to keep the prone element of the Yo-Yo IR1 test due to its increased validity to simulated rugby league match play but reduce the distance from 20m to 15m. From a practical perspective, the reduced distance increased the sensitivity of the testing measure facilitating a greater comparison of fitness levels. Whilst the authors appreciate that the 15m prone Yo-Yo has not been specifically validated via previous literature, we believe that the similarity in this protocol to the validated 20 prone Yo-Yo offers an appropriate measure of aerobic capacity and has been included within the testing battery. 

• Dobbin, N., Highton, J., Moss, S. L., Hunwicks, R., & Twist, C. (2021). Concurrent validity of a rugby-specific Yo-Yo intermittent recovery test (level 1) for assessing match-related running performance. The Journal of Strength & Conditioning Research, 35(1), 176-182.

The results and discussion will be reviewed in the 2nd revision based on the feedback provided on the Yo-Yo test that tested the aerobic capacity.

---

## [Editor Report · Decision Letter 1]

12 Jan 2022

The anthropometric and physical qualities of women’s rugby league Super League and international players; identifying differences in playing position and level

PONE-D-21-08836R1

Dear Dr. Scantlebury,

We’re pleased to inform you that your manuscript has been judged scientifically suitable for publication and will be formally accepted for publication once it meets all outstanding technical requirements.

Kind regards,

Caroline Sunderland

Academic Editor

PLOS ONE
---

## [Editor Report · Acceptance letter]

21 Jan 2022

PONE-D-21-08836R1 

The anthropometric and physical qualities of women’s rugby league Super League and international players; identifying differences in playing position and level 

Dear Dr. Scantlebury:

I'm pleased to inform you that your manuscript has been deemed suitable for publication in PLOS ONE. Congratulations! Your manuscript is now with our production department. 

Kind regards, 

on behalf of

Dr. Caroline Sunderland 

Academic Editor

PLOS ONE